# Novel Treatment Strategy for Management of Traumatic Bulbar Urethral Rupture Using Temporary Urethral Stent after Primary Realignment; Retrospective Comparison between Thermo-Expandable Urethral Stent and Self-Expandable Polymer-Coated Urethral Stent

**DOI:** 10.3390/jcm9051274

**Published:** 2020-04-28

**Authors:** Sun Tae Ahn, Dong Hyun Lee, Jong Wook Kim, Du Geon Moon

**Affiliations:** Department of Urology, Korea University Guro Hospital, No. 148, Gurodong-ro, Guro-gu, Seoul 08308, Korea; asturology@gmail.com (S.T.A.); donghyunlee13@gmail.com (D.H.L.); kikko@hanmail.net (J.W.K.)

**Keywords:** urethra, trauma, rupture, urethral realignment, urethral stenting

## Abstract

A variety of retrievable and other types of temporarily placed stents are currently being used. However, only a few studies have considered primary endoscopic realignment with temporary urethral stent insertion in the event of traumatic bulbar urethral injury. We aimed to compare the clinical effectiveness and complications between thermo-expandable urethral stents and polymer-coated bulbar urethral stents (BUSs) for the treatment of traumatic bulbar urethral strictures. Between September 2011 and March 2018, 30 patients who had been diagnosed with complete bulbar urethral rupture following blunt trauma underwent temporary urethral stent placement after primary realignment. Thermo-expandable nickel-titanium alloy urethral stents were placed for 15 patients (group M), and retrievable self-expandable polymer-coated BUSs were placed for another 15 patients (group A). All stents were removed within 6 months after placement. The complications and maintained patency rates were compared between the two groups. The mean stent indwelling period was 5.0 ± 2.5 months in group M and 4.9 ± 4.0 months in group A. Both groups maintained high patency rates (Group M 12/15 (80.0%) and group A 13/15 (86.7%)). Five patients who developed urethral stricture underwent direct visual internal urethrotomy (DVIU), and no patients required repeat DVIU or open surgical urethroplasty. Both groups maintained the mean maximal urinary flow rate (Qmax) at 12 months after stent removal. Discomfort (46.7% vs. 6.7%), granulation tissue formation (73.3% vs. 26.7%) and post-void dribbling (80.0% vs. 20.0%) were more frequent in group M than in group A (*p* = 0.013, *p* = 0.011 and *p* = 0.001, respectively). In conclusion, both stents were effective for managing traumatic complete bulbar urethral rupture after primary realignment. However, the thermo-expandable urethral stents had a higher complication rate while the stent was in situ than the BUSs.

## 1. Introduction

The bulbar urethra is the most common injury site in the anterior urethra for non-iatrogenic trauma [1,2]. The majority of injuries are caused by blunt trauma, mostly including ‘straddle injuries’, vehicular accidents, or kicks to the perineum [3]. In these bulbar injuries, the urethra may be either partially or completely disrupted, resulting in the loss of urethral continuity, local bleeding, urinary extravasation, inflammation and scarring. The optimal treatment strategy for bulbar urethral injury remains controversial [4,5,6]. Traditionally, immediate suprapubic cystostomy followed by delayed urethroplasty is preferred if the stricture has been confirmed [7,8]. However, long-term suprapubic tube drainage is associated with complications, including wound infection, urinary tract infection, catheter encrustation, urinary leakage, patient discomfort and inconvenience in performing daily activities that can decrease the patient’s quality of life [9]. 

Recent advances in endoscopic techniques have seen early realignment and transurethral catheterization of the injured segment introduced as new management options; consequently, recent clinical guidelines have listed primary endoscopic realignment as a treatment option for traumatic bulbar urethral injury. The aim of realignment is to allow the urethral injury to heal in the correct position rather than to prevent a stricture. In previous studies, the stricture formation rate was reported as 75–100% in cases with complete rupture [1,2].

The primary endoscopic realignment must be improved to reduce the stricture rate while maintaining the advantages of endoscopic realignment, especially in cases of complete bulbar urethral rupture. Thus, a new surgical technique, determining optimal catheter indwelling time or requirement of additional procedures using devices, is warranted. Recent developments in mechanical devices have led to the development of new types of temporary urethral stents, including thermo-expandable urethral stents and polymer-coated bulbar urethral stents (BUSs); these have demonstrated favorable therapeutic effects in recurrent urethral strictures [10,11,12]. However, there are limited reports regarding primary endoscopic realignment with temporary urethral stent insertion in traumatic urethral injury cases [13]. Additionally, comparative studies of the efficacy and safety of different types of stents are lacking. Therefore, this study reports our experience of primary repair with temporary urethral stent insertion in complete traumatic bulbar urethral rupture. We also compare the clinical effectiveness and complications between thermo-expandable urethral stents and polymer-coated BUSs for the treatment of traumatic bulbar urethral injury.

## 2. Materials and Methods

### 2.1. Patients and Study Design

A retrospective study was performed for patients who underwent primary realignment and temporary urethral stent insertion for bulbar urethral rupture. Institutional review board (IRB) approval was obtained for this study (IRB No. 2020GR0006), and informed consent was waived because of its retrospective nature. 

We identified 36 patients who had suffered complete bulbar urethral rupture caused by blunt trauma between September 2011 and March 2018. All the patients were diagnosed with urethral rupture by retrograde urethrography. Complete urethral rupture was defined as the extravasation of a contrast agent at the injury site with no contrast entering the prostatic urethra or the bladder. Patients who had concomitant pelvic bone fracture or other organ injury, with the exception of scrotal and perineal hematomas, were excluded from the study. Patients who received attempts at catheterization before retrograde urethrography were also excluded, as were patients with a follow-up duration of less than 1 year after stent removal. Finally, a total of 30 patients were analyzed in this study. 

Patients were divided into two groups according to the type of temporary stent. The patients who had received thermo-expandable stents were defined as group M, and the patients who had received a self-expandable stent were defined as group A. The baseline characteristics, including age, urethral defect length and stent indwelling duration were compared between the two groups.

### 2.2. Stent

Two types of stents were used in this study. Thermo-expandable urethral stents (Memokath 044TW, Pnn Medical, Kvistgaard, Denmark) were used in group M, and self-expandable stents (Allium Bulbar Urethral Stent, Allium LTD, Caesarea, Israel) were used in group A. 

The Memokath stent is made of a tightly coiled, nickel-titanium alloy that is designed to prevent urothelial ingrowth. It has a thermosensitive shape memory that expands either at the proximal segment or at both ends of the stent from 24 to 42 Fr, using warm water (55 °C) instillation to facilitate the anchoring of the stent in the correct position. When cooled irrigant is used (approximately 5–101 °C), the expanding ends become soft and pliable and allow for easy removal. The stents range in length from 3 to 8 cm in 1 cm increments; this allows selection of the appropriate stent length for individual patients.

The Allium bulbar urethral stent (BUS) is a fully covered, large caliber metal stent specially designed for the treatment of bulbar urethral strictures (Figure 1). The stent is built of a coiled, superelastic metal alloy (nitinol) covered with a polymeric coating designed to prevent mucosal hyperplasia and encrustations. The main body (Figure 1B) acts as a mold to allow formation of a large urethral lumen expanding to 45 Fr caliber. The dynamic sphincteric segment (Figure 1A) prevents sphincteric dysfunction, which may lead to incontinence. The last portion of the stent is the soft distal segment (Figure 1C). A special unraveling feature allows stent retrieval by unraveling it into a thread-like strip and enables a nontraumatic removal. The stent is available in three different lengths: 50, 55, and 60 mm.

### 2.3. Management Strategy 

At the time of trauma, all patients were diagnosed with a urethral rupture by retrograde urethrography. Once the complete urethral rupture was diagnosed, patients were treated primarily with suprapubic cystostomy for urinary diversion and perineal compression for reducing perineal hematoma. 

After a week, primary endoscopic realignment was performed under general or spinal anesthesia. With the patient in the lithotomy position, the suprapubic percutaneous tract was dilated, and a flexible cystoscope was passed through the bladder neck into the prostatic urethra. At this point, a 20-Fr urethrotome, with its half sheath, was carefully advanced to meet the tip of the light of the flexible cystoscope. Under endoscopic guidance, the urethrotome was further advanced into the prostatic urethra, while the flexible cystoscope was retracted to the bladder neck. The urethrotome was then removed, leaving the half sheath in the bladder and urethra. Through the half sheath, an 18-Fr Foley catheter was placed into the bladder. 

A month after the trauma, either a thermo-expandable nickel-titanium alloy urethral stent or a self-expandable polymer-coated BUS was placed temporarily. The patients were placed in the lithotomy position with spinal or general anesthesia. The previous rupture site was examined to ascertain whether it was sufficiently healed and whether spongio-fibrosis was yet to set in, under direct vision of a urethroscope. The urethral defect length was measured during surgery and defined as the injured longitudinal length to be healed. Subsequently, stents were deployed with specially designed delivery systems to cover the previous rupture site by over 1 cm length at both ends. No Foley catheter was inserted after the urethral stent placement. Patients were advised to visit the outpatient clinic every 1 or 2 months until the stent was removed.

All stents were planned to be removed within 6 months. The patients were placed in the lithotomy position, and the stent removal procedures were performed under local or general anesthesia according to the patients’ preferences. After stent removal, we explored the previous rupture site under urethroscope. There were no wound contractions and circumferential scar formations that resulted in stricture at the previous rupture site (Figure 2). 

All patients were instructed to return for follow-up at 1, 3, 6 and 12 months after stent removal. During the initial follow-up visits at 1 and 3 months after stent removal, all patients who showed urethral overgrowths at the edge of the insertion site of the stent underwent urethral dilatation. During the follow-up, no self-dilatation was permitted. The summarized treatment strategy algorithm is presented in Figure 3.

### 2.4. Assessments

The clinical success rate and complications were compared in groups M and A during both the stent indwelling period and after stent removal.

All patients were assessed with uroflowmetry and plain radiography during the stent indwelling period, and stent-related symptoms were assessed every visit until the stent removal. Stent-related complications, including migration, discomfort, tissue ingrowth, granulation at stent edge, stone formation and post-void dribbling, were recorded and compared between groups. 

After stent removal, all patients were assessed with uroflowmetry and urethroscopy at 1, 3, 6 and 12 months after stent removal. Clinical success was defined as maintenance of urethral patency in urethroscopy, passing of a 17-Fr cystoscope without resistance and a maximum urinary flow rate > 10 mL/s without requirement for additional surgery such as visual internal urethrotomy or urethroplasty. 

The Mann–Whitney U test was used to compare groups for continuous variables, and the Fisher exact test was used to compare categorical variables. A two-sided *p*-value < 0.05 was considered statistically significant for all tests. All analyses were performed using IBM SPSS Statistics version 22.0 (IBM Co., Armonk, NY, USA).

## 3. Results

Of the 30 patients selected for this study, 15 were treated with thermo-expandable stents and were classified as group M, while the other 15 patients were treated with self-expandable stents and were classified as group A. The demographic and clinical characteristics of both groups are presented in Table 1. There were no significant differences between groups M and A in terms of mean patient age, urethral defect length, and stent indwelling duration.

The maintained urethral patency rate at 1 year after stent removal was 80.0% (12/15) in group M and 86.7% (13/15) in group A, with no significant difference between the groups. Uroflowmetry showed that both groups maintained mean Qmax to 1 year after stent removal (Figure 4). Five (16.7%) of 30 patients, three patients in group M and two patients in group A, failed to maintain urethral patency after stent removal. Five patients who developed urethral stricture underwent direct visual internal urethrotomy (DVIU), and no patients required repeat DVIU or open surgical urethroplasty. 

The complications observed during stent indwelling are detailed in Table 2. Discomfort was significantly more frequent in group M (7/15, 46.7%) than in group A (1/15, 6.7%) (*p* = 0.013). All incidences of stent-related discomfort were managed with analgesics, and no one required stent removal. The granulation tissue formation rate at the stent edge differed significantly (*p* = 0.011) between groups, with 11 patients in group M and 4 patients in group A. Tissue granulation at the stent edge was managed by repeat urethral dilatation during the first 3 months after stent removal. One patient in group M required transurethral resection of granulation tissue due to sustained obstructive granulation tissue even after repeated urethral dilatations. Post-void dribbling was the most frequent complication in group M (12/15, 80.0%), while only three patients (3/15, 20.0%) in group A demonstrated symptoms (*p* = 0.001); post-void dribbling was controlled by Kegel exercises with or without an anticholinergic drug. Other complications, including stent migration and stent encrustation, did not differ significantly between the groups. Stent encrustations were not obstructive and were managed by stent removal before obstruction. Stent migration was managed with stent re-positioning by urethroscopy in group A, while three patients in group M underwent early stent removal. Although stent removal was done principally under local anesthesia, general anesthesia was performed on six patients with tissue granulation, three patients with encrustation and three patients by request. No difficulty in stent removal was noted in either stent, even in those with encrustation, migration or tissue granulation issues. 

## 4. Discussion

Numerous studies have reported the efficacy of temporary urethral stent insertion for the management of urethral stricture [10,11,14,15]. However, most of the studies have focused on recurrent urethral strictures. No previous study has been conducted to determine the efficacy of temporary urethral stent placement after primary realignment for traumatic bulbar urethral ruptures. In the current study, we first identified a treatment strategy for bulbar urethral rupture with temporary urethral stent placement after primary realignment. We also analyzed the urethral patency rate and the concomitant complications of two different types of urethral stents. We confirmed that both urethral stents showed a sustained Qmax 1 year after stent removal. However, self-expandable stents were shown to be preferable in terms of stent-related complications, including post-void dribbling, granulation tissue formation and discomfort. 

In this study, we enrolled patients with complete bulbar urethral rupture but not partial urethral rupture, since cases of partial urethral rupture usually heal rapidly. According to previous studies, the rate of stricture formation after primary realignment or suprapubic cystostomy alone was reported as 10–60%, and the stricture segments were generally short [1,2,16]. On the other hand, previous studies have also indicated that complete urethral rupture results in 100% stricture formation [1,2]. Therefore, additional treatment is needed to prevent stricture formation after primary realignment, especially in cases of complete bulbar urethral rupture.

According to a previous study, immediate primary surgical reconstruction is technically difficult due to the presence of acutely inflamed tissue, hematoma and anatomical distortion [17]. Therefore, the first step for our treatment strategy was perineal compression rather than immediate primary realignment. As the hematoma and urethral debris made it difficult to find the proximal urethral way, we treated with perineal compression for 1 week before performing primary realignment; with this strategy, there was no failure of realignment. The next main strategy was insertion of a temporary urethral stent before the urethral stricture settled. According to previous studies, most urethral strictures occurred or were reported at least a month after primary realignment. Based on this, we placed a temporary urethral stent within one month of injury. Using the above treatment strategy, we obtained a high overall urethral patency of more than 80% and demonstrable efficacy in preventing stricture. It is assumed that a stent functions as a scaffold in preventing wound contraction due to collagen deposition during wound healing of the urethra [11]. Eventually, when epithelization occurs at the rupture site, it appears to act as a mechanical force on the scar contraction to maintain a stable luminal patency.

The main drawback of our treatment strategy was that it required at least three endoscopic procedures, including primary realignment, temporary urethral stent insertion and stent removal. This was due to the necessity of performing temporary urethral stent insertion to prevent stricture formation and maintain urethral patency after primary realignment in complete bulbar urethral injury. However, the main objective of our treatment strategy was to minimize urethral stricture while maintaining primary realignment advantages, including shortening the time to spontaneous voiding and reducing the morbidity associated with long-term suprapubic cystostomy in complete bulbar urethral rupture. Although initial suprapubic cystostomy with delayed urethroplasty reported better outcomes in complete bulbar urethral rupture [2], massive distraction of urethral ends often resulted in complicated and lengthy strictures which required complex flap or grafting surgery [18]. The clinician should inform the patients about the risks and benefits of each treatment, including cost, inconvenience and complications before starting the treatment. 

Another consideration was stent-related complications and patient discomfort during the period when the urethral stent was in situ. One of the reasons that primary realignment treatment is preferable is related to the short suprapubic diversion time, but it is paradoxical to see a decrease in quality of life (QOL) due to the indwelling urethral stent, in patients who undergo temporary urethral stent insertion after primary realignment. Unfortunately, this study did not directly compare delayed repair after suprapubic cystostomy and complications that affect QOL. Instead, we investigated the complications of two popular temporary urethral stents and found that BUSs have a lower incidence of stent-related complications than the Memokaths; this seems to be primarily related to the structural difference between the two stents. The BUS has a body with large caliber of a hard material, that provides a wide patent lumen, and two soft ends [19], while Memokath expands and is fixed at both ends. As a result, the structural differences in the ends of the stents appears to lead to the formation of tissue granulation at the edge of the stents and patient discomfort. Additionally, the proximal soft end of the BUS faces the proximal urethra/sphincter, which has the advantage of preventing post-void dribbling or incontinence.

The placing of temporary urethral stents for the management of urethral ruptures has not yet been studied, and as a result the optimal duration of stent dwell-time has not been determined. For ideal treatment, the stent should remain in the rupture site until complete recovery but should be removed before invasion of granulation tissue at the end of the stent. Although there is no established strategy for stent removal, the stricture length, stent-related discomfort, degree of urethral disruption and underlying disease should all be considered. We also believe that the implanted stent will serve as a scaffold during the wound-healing phase and over the duration of maximal support, while its ability to reduce complications will vary from patient to patient; thus, the optimal duration of stent dwell-time should be considered on a case-by-case basis.

Our study has several limitations. Firstly, we could only analyze data from a small number of patients due to the rarity of complete bulbar urethral rupture from trauma. Secondly, because the study was not conducted prospectively, randomization could not be achieved, and the study results may have been affected by selection bias. However, in order to minimize the selection bias, we only included patients with complete bulbar urethral rupture. Thirdly, we did not compare our treatment strategy to traditional open urethroplasty. However, the aim of our study was primarily focused on primary endoscopic realignment to confirm meaningful results in reducing stricture formation through temporary urethral stent insertion in complete bulbar urethral rupture. Fourthly, sexual dysfunction was not evaluated in this study. Due to retrospective design, the baseline sexual function was unknown in this study’s cohort and could not be evaluated appropriately. A final potential limitation of this study was the relatively short duration of follow-up. However, in previous studies the majority of strictures were reported within 3 months of temporary stent removal [1,16]; as a result, we believe that the 1 year follow-up in the current study was sufficient to demonstrate the efficacy of the stents. Despite these limitations, we believe this study prompts further investigation to improve minimally invasive treatments in traumatic urethral ruptures.

## 5. Conclusions

Urethral stents have long been used to preserve luminal patency; with the development of endoscopic procedures, these stents have also been further developed for various purposes. While the majority of previous studies have focused on recurrent urethral strictures, in this study, we investigated temporary urethral stents after primary realignment in traumatic bulbar urethral ruptures. Both stents were effective in preserving urethral patency after complete bulbar urethral rupture. Self-expandable BUSs showed fewer complications compared to thermo-expandable urethral stents. The temporary urethral stent may play a role in the endoscopic management of complete bulbar urethral rupture.

## Figures and Tables

**Figure 1 jcm-09-01274-f001:**
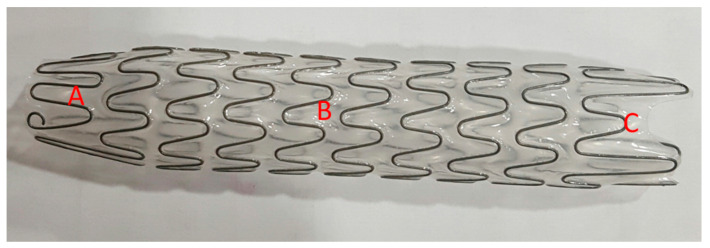
An Allium bulbar urethral stent (BUS). (**A**) Soft sphincteric segment; (**B**) high radial force body; (**C**) soft distal segment.

**Figure 2 jcm-09-01274-f002:**
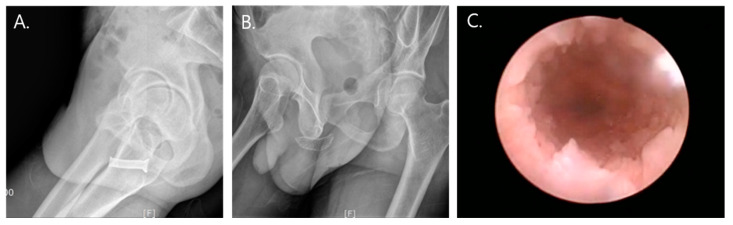
(**A**,**B**) Temporary urethral stents indwelled over the previous rupture site (A: Memokath, B: Allium bulbar urethral stent). (**C**) Urethroscopy after stent removal.

**Figure 3 jcm-09-01274-f003:**
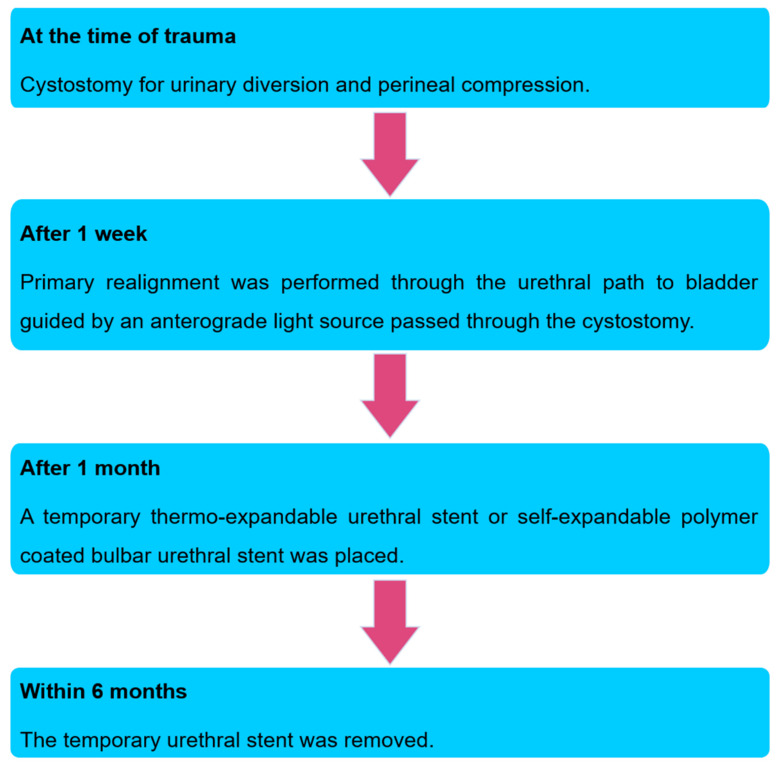
Treatment strategy algorithm.

**Figure 4 jcm-09-01274-f004:**
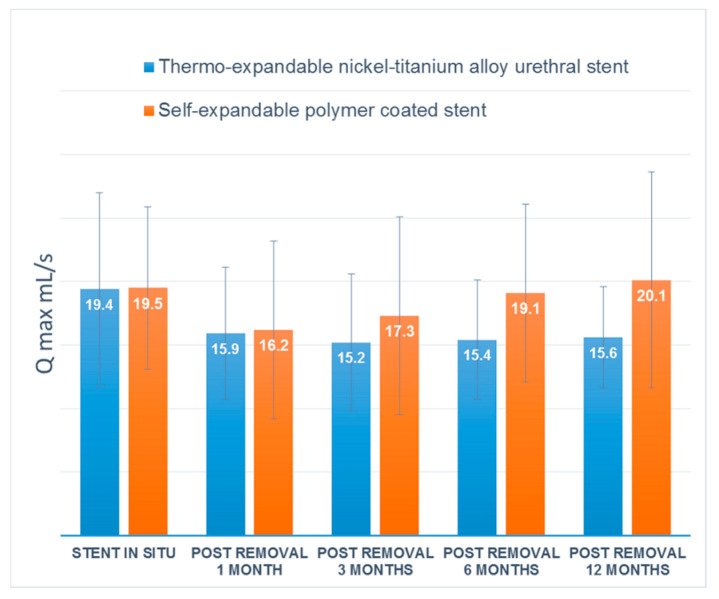
Mean maximal urinary flow rates (Qmax) after stent removal.

**Table 1 jcm-09-01274-t001:** A summary of baseline clinical characteristics of patients in both groups.

Patient Characteristics	Group M * (*n* = 15)	Group A ** (*n* = 15)	*p*-Value
Mean age (years)	50.9 ± 13.3	53.9 ± 9.6	0.494
Urethral defect length (cm)	3.1 ± 0.7	3.0 ± 1.2	0.858
Stent size (cm)	4.2 (3.0–6.0)	6.0	-
Stent indwelling period (months)	5.0 ± 2.5	4.9 ± 4.0	0.957

Values are expressed as the mean ± SD; * Patients who were treated with thermo-expandable stents; ** Patients who were treated with self-expandable stents.

**Table 2 jcm-09-01274-t002:** Stent-related complications in both groups.

Complication	Group M (*n* = 15)	Group A (*n* = 15)	*p*-Value
Pain	7 (46.7%)	1 (6.7%)	0.013
Stent Encrustation	2 (13.3%)	2 (13.3%)	-
Stent migration	3 (20.0%)	2 (13.3%)	0.624
Tissue granulation at stent edge	11 (73.3%)	4 (26.7%)	0.011
Post-void dribbling	12 (80.0%)	3 (20.0%)	0.001

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
