# Peer review of "Novel Treatment Strategy for Management of Traumatic Bulbar Urethral Rupture Using Temporary Urethral Stent after Primary Realignment; Retrospective Comparison between Thermo-Expandable Urethral Stent and Self-Expandable Polymer-Coated Urethral Stent"

_jcm, 2020, doi:10.3390/jcm9051274_

Round 1
Reviewer 1 Report
the authors present their experience on the use of expandable stents for the temporary treatment of complete post traumatic bulbar urethral rupture.
the study design is original and the patient series is homogeneous. despite no randomization, the two groups treated with two different self expandable stents are similar. comparison of results is strict and statistics correctly used. followup could be longer but authors recognize it as one of the limits of the study. the discussion is rich, conclusions are sound and references are complete. Limits and strenght of the study are well indicated
little editing mistakes
line 79 urethral instead of ureteral
line 197-199 need be cancelled
line 233 ironic?
Author Response
little editing mistakes
Point 1: line 79 urethral instead of ureteral
Response 1: As you point out, we revised the word
Point 2: line 197-199 need be cancelled :
Response 2: As you point out, we cancelled the sentences.
Point 3: line 233 ironic?
Response 3: As you pointed out, we corrected the sentence to clarify the meaning.

Reviewer 2 Report
Study found that with complete bulb are urethral disruption patent see rates after temporary urethral stinting at 1 year will greater than 80%. It appears patients tolerated the polymer coated stent the best. I think this is a great contribution to our literature and will be a catalyst for further investigation how to advance minimally invasive urethral surgery.
- My biggest concern is the number of procedures required from initial trauma up to 6 months after urethral stent removal. I see that there were at least 6 endoscopic procedures of which at least 4 required anesthesia. As we emphasize, cost, inconvenience and risk to the patient this is not insignificant. If the patient were to have a supra pubic tube placed with delayed urethra plastic this only requires 2 procedures with a very high >90% success rate. I think the authors need to expand on this in the discussion.
- I also have concern with the granulation tissue that formed along the edges of the stent, it seems that this is a precursor to delayed urethral stricture formation. I am also concerned that the patient's underwent several urethral dilations after stent removal. Can the authors clarify, was this only done at the 1 month and three-month interval in clinic? Or were these patients performing intermittent self dilation continuously. Furthermore I see that the performance of self dilation was not considered a treatment failure, were many of these patients continuing to perform self dilation up to 1 year after stent removal?
- How many patients required the stent removal under general anesthesia? Was there difficulty with stent removal in any these patients either due to migration, encrustation, or tissue ingrowth?
- What was the location of the urethral injury? Did it occur in the distal, mid or proximal bulbar urethra. I can see that this may predispose patients to increasing pain with distal injuries and predispose patients to increase post void dribbling or urinary leakage with proximal injuries
- Was the rationale to use a thermo expandable verses polymer stent?
- With sexual dysfunction assessed? if so was there a difference between the 2 cohorts
Author Response
Point 1: My biggest concern is the number of procedures required from initial trauma up to 6 months after urethral stent removal. I see that there were at least 6 endoscopic procedures of which at least 4 required anesthesia. As we emphasize, cost, inconvenience and risk to the patient this is not insignificant. If the patient were to have a supra pubic tube placed with delayed urethra plastic this only requires 2 procedures with a very high >90% success rate. I think the authors need to expand on this in the discussion.
Response 1: Thank you for your comment. You pointed out the drawbacks of our treatment strategy. Multiple endoscopic procedures were due to the necessity of performing temporary urethral stent insertion to prevent stricture formation after primary realignment in complete bulbar urethral injury. We agree with your comments and added the disadvantages of the procedure to the discussion.
Point 2: I also have concern with the granulation tissue that formed along the edges of the stent, it seems that this is a precursor to delayed urethral stricture formation. I am also concerned that the patient's underwent several urethral dilations after stent removal. Can the authors clarify, was this only done at the 1 month and three-month interval in clinic? Or were these patients performing intermittent self dilation continuously. Furthermore I see that the performance of self dilation was not considered a treatment failure, were many of these patients continuing to perform self dilation up to 1 year after stent removal? :
Response 2: Thank you for your comment. Urethral dilatation was performed by the authors at 1 and 3 months for all patients who had seen urethral overgrowth at the edge of the insertion site of the previous stents. Patients who showed sustained obstructive granulation tissue after 3 months were considered as treatment failure and 1 patient had this finding as mentioned in Result section. Self-dilatation was not permitted. We added and clarified the above information to Material methods.
Point 3: How many patients required the stent removal under general anesthesia? Was there difficulty with stent removal in any these patients either due to migration, encrustation, or tissue ingrowth?
Response 3: Thank you for your comment. Although stent removal procedure was done principally under local anesthesia, the general anesthesia was performed on 6 patients with tissue granulation, 3 patients with encrustation and 3 patients by demand. Stone formation or migration had no difficulty in removing the stent. The above information was added to the result.
Point 4: What was the location of the urethral injury? Did it occur in the distal, mid or proximal bulbar urethra. I can see that this may predispose patients to increasing pain with distal injuries and predispose patients to increase post void dribbling or urinary leakage with proximal injuries
Response 4: Thank you for your comment. All patients were injured from blunt perineal trauma including straddle injury, thus the study subjects were homogenous series with mid to proximal bulbar urethral injury. However, the stents were covered over the rupture site including distal bulbar urethra, so we considered that the incidence of stents related complications were primarily related to the structural differences between stents.
Point 5: Was the rationale to use a thermo expandable verses polymer stent?
Response 5: Before 2016, self-expandable polymer coated stent was not available in Korea. Thus, we used thermo-expandable stent from 2011 to 2015.
Point 6: With sexual dysfunction assessed? if so was there a difference between the 2 cohorts
Response 6: Unfortunately, the sexual dysfunction could not evaluated in this study due to retrospective design. In future prospective studies, it would be consider evaluating the postoperative complications for sexual dysfunction, as you suggested. The above limitation was added to the discussion.

Reviewer 3 Report
This is a completely underpowered study. A comparative study should be randomized.The new technique must be compared with the standard treatment which is in this situation an end-to-end anastomosis or substitution urethral reconstruction. The results described in your study are of the same success rate as those described with this standard treatment. This study doesn't allow any conclusion.
Author Response
Point 1: This is a completely underpowered study. A comparative study should be randomized.The new technique must be compared with the standard treatment which is in this situation an end-to-end anastomosis or substitution urethral reconstruction. The results described in your study are of the same success rate as those described with this standard treatment. This study doesn't allow any conclusion
Response 1: Thank you for your comment. We agree with your opinion. Our study has limitation that the study did not compare with delayed open urethroplasty. We have mentioned as a limitation in discussion section of the revised manuscript. However, the aim of our study was primarily focused on primary endoscopic realignment to confirm meaningful results in reducing stricture formation through temporary urethral stent insertion in complete bulbar urethral rupture. We believe our study would drive further investigation to improve minimal invasive treatment in traumatic urethral injury.

Round 2
Reviewer 2 Report
ok to accept with provided changes